# Synergistic Activity of Dimethyl Disulfide Mixtures with Two Chemical Compounds against *Meloidogyne incognita*

Qing Wang, Lirui Ren, Daqi Zhang, Zhaoxin Song, Wensheng Fang, Yuan Li, Qiuxia Wang, Aocheng Cao and Dongdong Yan *

Institute of Plant Protection, Chinese Academy of Agricultural Sciences, Beijing 100193, China
* Correspondence: yandongdong@caas.cn

**Abstract:** The prevention and control of root-knot nematode disease is a worldwide challenge and there are not many varieties of pesticides for nematode control. To meet the huge market demand, the development of new nematicides is urgently needed. The lethal effects of soil fumigant dimethyl disulfide (DMDS) mixed with the chemical compounds copper sulfate ($CuSO_4$) and ammonium bicarbonate ($NH_4HCO_3$) on *Meloidogyne incognita* were tested using the immersion method. The results showed that the $LC_{50}$ of DMDS, $CuSO_4$, and $NH_4HCO_3$ on the second stage juveniles (J2) of *M. incognita* were 19.28, 187.42, and 213.49 mg/L, respectively. The lethal effect on J2 were enhanced with the combination of DMDS and $CuSO_4$ or $NH_4HCO_3$. The compound uses of DMDS (2.5 mg/L) and $CuSO_4$ (46.58 mg/L) or $NH_4HCO_3$ (80.25 mg/L) have obvious synergistic effects on the control of *M. incognita*, with corrected mortalities of 97.09% and 94.00%, respectively. The synergistic effect of fumigant and chemical compounds on *M. incognita* was investigated to provide a new concept for the control of root-knot nematode disease.

**Keywords:** dimethyl disulfide; copper sulfate; ammonium bicarbonate; *Meloidogyne incognita*





## 1. Introduction

Plant-parasitic nematodes (PPN) are among the main invasive pathogens of plants. PPN are distributed worldwide and cause damage to important crops. More than 100 species of nematodes cause serious harm to agriculture, forestry, and economically important crops in China. The damage caused by PPN to some crops even exceeds that of other diseases, insects, and weeds [1], causing approximately USD 157 billion in losses to global agriculture every year [2], of which USD 10 billion is caused by *Meloidogyne* spp. [3]. *Meloidogyne* spp. (root-knot nematodes, RKN) are considered the most destructive species of PPN in the world and include four major species, namely, *M. arenaria*, *M. hapla*, *M. incognita*, and *M. javanica* [4]. Nematode control is subject to integrated pest management (IPM) [5,6].

Several management practices can be used to control nematode diseases. These include cultural, chemical, and biological controls and host resistance. The effective management of PPN has relied upon the application of chemical nematicides as a short-term control means by suppressing nematode population densities in soil to levels below known economic damage thresholds [7,8]. Chemical nematicides have been used singly or in combination with other nematode control practices since the late 19th century [9]. Nematicides can be classified into fumigants and non-fumigants, according to their action modes. Fumigant nematicides are applied as a liquid or gas and generate lethal volatiles that diffuse as gas into the soil and kill the nematodes [10]. Most fumigant nematicides are currently banned in many countries over concerns about their impacts on human health and the environment. For example, the fumigant methyl bromide damages the atmospheric ozone layer [11]. Others, such as dazomet, meta sodium, chloropicrin, 1,3-dichloropropene, sulfuryl fluoride, and dimethyl disulfide are still in use for controlling nematode diseases [12]. Other non-fumigant, but highly toxic, nematicides, such as aldicarb [13] and carbopol [14], are banned

because of environmental problems. Only 10 nematicides are commonly used, although more than 30 active ingredients are known [1]. Therefore, the chemical control strategies of PPN face severe barriers in their application.

Focusing on the development of new environmentally friendly pesticides has become a research focus in the control of RKN disease in recent years. The scientific use of organic and inorganic compounds have a certain control effect on RKN [15]. Humic acid and trace elements such as Fe, Mn, and Cu have obvious inhibitory effects on the survival of *M. incognita* second-stage juveniles (J2) [16]. N, P, Cu, Mn, and K at very high concentrations have significant mortality rates on the *Heterodera avenae* J2 [17].

In a world where food safety and environmental concerns are increasing, assessing the compatibility of mixtures with other crop protection products and compounds to minimize the number of required applications is an important task. Therefore, the toxicity of the soil fumigant dimethyl disulfide (DMDS) combined with chemical compounds copper sulfate ($CuSO_4$) and ammonium bicarbonate ($NH_4HCO_3$) to *M. incognita* was evaluated to provide a novel concept for the prevention and control of RKN.

## 2. Materials and Methods

### 2.1. Nematode Source

The nematodes were collected from infected tomato roots (*Solanum lycopersicum* L., Variety Provence) in Nanhe village, Dashiwo Town, Fangshan District, Beijing ($115°48'2''$ E, $39°31'33''$) and identified as *M. incognita* via morphological and molecular identification. The root system was cleaned, single females were isolated and perineal patterns were prepared and observed, according to the methods of Zhang et al. [18] and Feng [19]. The DNA of J2 was extracted according to the method of Feng et al. [20]. A specific primer (MI-F: 5′-GTGAGGATTCAGCTCCCCAG-3′, MI-R: 5′-ACGAGGAACATACTTCTCCC-3′) [21] was used for PCR amplification. This primer was designed based on the specific RAPD fragments OP26-01$_{1200}$, which amplified a fragment of 995 bp from *M. incognita* [22]. Following the procedure of Hussey and Barker [23], fresh egg masses were picked from tomato roots. The eggs were extracted by applying 0.5% NaClO for 3 min, washed thrice with sterile water, and then placed into 24-well culture plates. The nematodes were incubated at room temperature (25 °C) to promote the hatching of J2. The freshly hatched J2 were collected every day and stored at 4 °C until further use.

### 2.2. Test Compounds

Dimethyl disulfide (98%, Shanghai Maclean Biochemical Technology Co., Ltd., Shanghai, China), copper sulfate ($CuSO_4·5H_2O$, Analytical Purity, Sinopharm Chemical Reagent Co., Ltd., Shanghai, China), and ammonium bicarbonate ($NH_4HCO_3$, 99%, Beijing Coolman Technology Co., Ltd., Beijing, China) were used as test compounds.

### 2.3. Determination of the Toxicity of DMDS, $CuSO_4$, and $NH_4HCO_3$ on M. incognita J2

In the immersion method, 0.5 mL of each test compound was added to 0.5 mL of nematode suspension (approximately 50 J2), resulting in final DMDS concentrations of 2.5, 5, 10, 20, 40, and 80 mg/L; final $CuSO_4$ concentrations of 11.65, 23.29, 46.58, 93.16, 186.33, and 372.66 mg/L; and final $NH_4HCO_3$ concentrations of 80.25, 160.5, 321, 642, 1284, and 2568 mg/L. The control treatment consisted of the 50 J2s maintained in 1 mL distilled water alone. Each treatment was replicated 6 times. The samples were maintained at 25 °C for 24 h. Approximately 0.5 mL of the mixture from each replicate was observed under a microscope and the number of total and dead nematodes was recorded. The nematodes were classified as dead when they were straight and did not move when touched with a hairpin.

### 2.4. Determination of Toxicity of DMDS with $CuSO_4$ or $NH_4HCO_3$ on M. incognita J2

The concentration gradients of DMDS, $CuSO_4$, and $NH_4HCO_3$ were diluted as follows. Approximately 0.5 mL of nematode suspension with 100 J2/mL was added to a 2 mL

centrifuge tube, followed by 0.5 mL of a mixture of DMDS with $CuSO_4$ or DMDS with $NH_4HCO_3$. The final concentrations of DMDS were 2.5, 10, and 20 mg/L. The final concentrations of $CuSO_4$ were 46.58 and 186.33 mg/L and those of $NH_4HCO_3$ were 80.25 and 241 mg/L. The samples were cultured at 25 °C for 24 h. Approximately 0.5 mL of the mixture from each replicate was observed under an anatomical microscope and the number of total and dead nematodes was recorded. The nematodes were classified as dead when they were straight and did not move when touched with a hairpin.

*2.5. Statistical Analysis of Data*

The data were used to calculate the corrected mortality using Formulas (1) and (2) [24]:

$$P = \frac{K}{N} \times 100 \tag{1}$$

where *P* is the mortality (%), *K* is the number of dead nematodes, and *N* is the number of total nematodes;

$$E = \frac{P_t - P_0}{1 - P_0} \times 100 \tag{2}$$

where *E* is the corrected mortality (%), $P_t$ is the mortality of treatments (%), and $P_0$ is the mortality of the controls (%).

The dose–response probabilistic model (PROBIT) was used to calculate the half-lethal median concentration $LC_{50}$ compound and its 95% confidence limit using the IBM SPSS Statistic V22.0.0 Software.

According to the Colby [25] method, the corresponding theoretical effect was calculated and compared with the real effect of the mixture. Formula (3) was used:

$$E - E_0 = E - X_1 X_2 / 100 \tag{3}$$

where *E* is the real measured efficacy of the combination, $E_0$ is the expected efficacy of the combination, $X_1$ is the real measured efficacy of the first compound, and $X_2$ is the real measured efficacy of the second compound. If $E > E_0$, the combinations were synergistic; if $E < E_0$, the combinations were antagonistic.

The experimental data on $E - E_0$ were subjected to one-way analysis of variance (ANOVA) and the means were compared using Duncan's multiple range test. The data in percentages were normalized with an arcsine square-root transformation.

**3. Results and Discussion**

*3.1. Lethal Effect of DMDS, $CuSO_4$, and $NH_4HCO_3$ on M. incognita J2*

The DMDS, $CuSO_4$, and $NH_4HCO_3$ had lethal effects on the *M. incognita* J2 and these effects increased with the concentration. The $LC_{50}$ on the J2 of *M. incognita* differed from each compound. After 24 h, the $LC_{50}$ values of DMDS, $CuSO_4$, and $NH_4HCO_3$ were 19.28, 187.42, and 213.49 mg/L, respectively (Table 1).

**Table 1.** Effects of DMDS, $CuSO_4$, and $NH_4HCO_3$ on *M. incognita* J2.

| Compound | Regression Equation | $LC_{50}$ (mg/L) | 95% Confidence Interval | Correlation Coefficient $R^2$ |
|---|---|---|---|---|
| DMDS | $y = 2.22x - 2.77$ | 19.28 | 17.25–21.61 | 0.93 |
| $CuSO_4$ | $y = 1.24x - 2.78$ | 187.42 | 155.03–234.84 | 0.85 |
| $NH_4HCO_3$ | $y = 1.57x - 3.66$ | 213.49 | 191.59–236.13 | 0.99 |

The nematode population densities were reduced after DMDS treatment in both indoor and field experiments. The $LC_{50}$ values reported for DMDS against *M. incognita* were 29.865, 0.086, and 6.348 mg/L depending on the application method, namely, the small tube method, desiccator fumigation, and soil fumigation, respectively [26]. In addition, DMDS has a good control effect against RKN on various crops. For example, DMDS

was used at 300 a.i. kg/ha in greenhouse experiments and obtained a control efficiency of 63–97% against the *M. incognita*, *M. javanica*, and *Heterodera schachtii* that occurred on tomatoes and sugar beets [27]. Low doses (112 kg/ha) of DMDS showed good effects on RKN during the grape growth period [28]. The DMDS acts on nematodes by affecting cytochrome oxidase in mitochondria [29]. Similarly, Dugravot et al. [30] found that DMDS reduces intracellular ATP concentration by inhibiting the mitochondrial respiratory chain complex IV (cytochrome oxidase) of *Periplaneta americana*. Subsequently, the activated neuronal $K_{ATP}$ channels mediate membrane hyperpolarization and decrease neuronal activity to control soil pests. Therefore, DMDS can show good activity against nematodes.

$CuSO_4$ is often used in agricultural production as a fungicide and micronutrient and, when used in moderate concentrations, it can replenish copper in the soil [31]. Moreover, trace elements such as Fe, Mn, and Cu significantly inhibit the survival of *M. incognita* J2 [16]. Similarly, $CuSO_4$ has a strong inhibitory effect on the survival of *M. incognita* J2 under in vitro conditions, thus significantly reducing their movement behavior, shortening their body length, and lengthening their transparent tails [32]. $CuSO_4$ has a strong toxic effect on *Bursaphelenchus xylophilus* and it is speculated that the killing mechanism may be through the combination of copper ions and proteins in the nematodes to form copper complexes. This process denatures and precipitates the proteins, causing enzyme inactivation, thereby hindering and inhibiting the metabolism and finally killing the nematode [33].

Ammonium nitrogen is a chemical compound that is often used in agricultural production and it is expected to become a nematicide [34,35]. The pot experiments of Oka and Pivonia [36] showed that among the 10 ammonium compounds tested, $NH_4OH$, $NH_4H_2PO_4$, and $NH_4HCO_3$ had significant nematicidal activity against *M. javanica*. The field experiments of Su et al. [37] showed that the total number of nematodes after treatment with $NH_4HCO_3$ combined with lime decreased by nearly 50% compared with $NH_4HCO_3$ alone and 66.2% compared with the control. The combination of lime bicarbonate and $NH_4HCO_3$ can exert a significant killing effect when the soil water content is low and in a wide temperature range. After application, the soil pH and the ammonium nitrogen, nitrate nitrogen, and total nitrogen contents can be significantly increased. The combined treatment of ammonium sulfate and alkaline-stable biosolids had a more significant effect than the single product. Although ammonium salts do not directly kill nematodes, they can form ammonia that are highly toxic to nematodes in alkaline soils [38]. Ammonia can be produced by organic matter with high nitrogen content and also by marine organisms with high chitin content, which can promote the dissolution of chitin on the surface of RKN and lead to the death of nematodes [39].

### 3.2. Lethal Effect of DMDS with $CuSO_4$ or $NH_4HCO_3$ on M. incognita J2

Based on the concentration of the single product, $CuSO_4$ and $NH_4HCO_3$ were mixed with DMDS (Table 2, Figure 1). The combinations of DMDS with $CuSO_4$ and $NH_4HCO_3$ enhanced the lethal effect on *M. incognita* J2. The expected efficacy of nematodes was 0.58–17.60% when DMDS was combined with $CuSO_4$. Actually, the corrected mortality of nematodes was more than 97% when combined with $CuSO_4$. The expected efficacy of nematodes was 1.15–18.32% when combined with $NH_4HCO_3$. However, the corrected mortality of nematodes was more than 95% when combined with $NH_4HCO_3$. When $E > E_0$, the combinations were all synergistic. In the combination of DMDS with $CuSO_4$, the effects of DMDS (2.5,10 mg/L) + $CuSO_4$ (46.58 mg/L) and DMDS (2.5 mg/L) + $CuSO_4$ (186.33 mg/L) were significantly different from the other concentrations. In the combination of DMDS with $NH_4HCO_3$, the effects of DMDS (2.5,10 mg/L) + $NH_4HCO_3$ (80.25, 241 mg/L) were significantly different from the other concentrations.

**Table 2.** Effects of DMDS combination with CuSO₄ or NH₄HCO₃ on *M. incognita* J2 mortality.

| Treatment | | | | Corrected Mortality (%) | | | |
|---|---|---|---|---|---|---|---|
| No. | Compound | Concentration (mg/L) | $E$ [a] | $E_0$ [b] | $E - E_0$ [c] | CE [d] | |
| 1 | DMDS | 2.5 | 4.42 | \ | \ | \ | |
| 2 | DMDS | 10 | 11.59 | \ | \ | \ | |
| 3 | DMDS | 20 | 35.13 | \ | \ | \ | |
| 4 | CuSO₄ | 46.58 | 13.02 | \ | \ | \ | |
| 5 | CuSO₄ | 186.33 | 50.11 | \ | \ | \ | |
| 6 | NH₄HCO₃ | 80.25 | 26.02 | \ | \ | \ | |
| 7 | NH₄HCO₃ | 241 | 52.15 | \ | \ | \ | |
| 8 | DMDS + CuSO₄ | 2.5 + 46.58 | 97.67 | 0.58 | 97.09 a | + | |
| 9 | DMDS + CuSO₄ | 10 + 46.58 | 98.46 | 1.51 | 96.95 a | + | |
| 10 | DMDS + CuSO₄ | 20 + 46.58 | 98.78 | 4.57 | 94.20 b | + | |
| 11 | DMDS + CuSO₄ | 2.5 + 186.33 | 98.99 | 2.21 | 96.78 a | + | |
| 12 | DMDS + CuSO₄ | 10 + 186.33 | 99.02 | 5.81 | 93.21 b | + | |
| 13 | DMDS + CuSO₄ | 20 + 186.33 | 100.00 | 17.60 | 82.40 c | + | |
| 14 | DMDS + NH₄HCO₃ | 2.5 + 80.25 | 95.15 | 1.15 | 94.00 a | + | |
| 15 | DMDS + NH₄HCO₃ | 10 + 80.25 | 95.50 | 3.02 | 92.48 a | + | |
| 16 | DMDS + NH₄HCO₃ | 20 + 80.25 | 95.55 | 9.14 | 86.41 b | + | |
| 17 | DMDS + NH₄HCO₃ | 2.5 + 241 | 96.06 | 2.30 | 93.76 a | + | |
| 18 | DMDS + NH₄HCO₃ | 10 + 241 | 99.63 | 6.04 | 93.59 a | + | |
| 19 | DMDS + NH₄HCO₃ | 20 + 241 | 100.00 | 18.32 | 81.68 c | + | |

[a]: $E$ = actual measured control efficacy of the combination. [b]: $E_0$ = expected control efficacy of the combination. [c] Different lowercases in the No.8–13 and No.14–19 indicated significant difference at 0.05 level by Duncan's multiple range test. [d] CE = combined efficacy; if $E - E_0 > 0$, CE was expressed as +; if $E - E_0 < 0$, CE was expressed as -; if $E - E_0 = 0$, CE was expressed as ±.

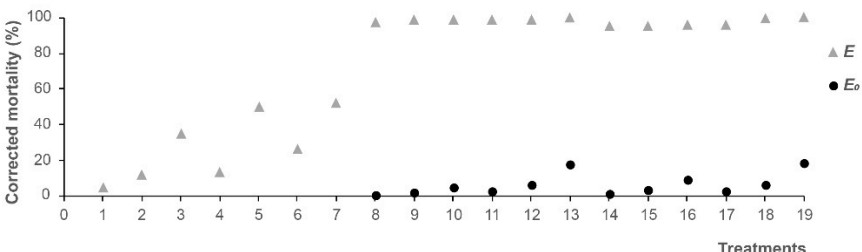

**Figure 1.** Effects of DMDS combination with CuSO₄ or NH₄HCO₃ on *M. incognita* J2 mortality.

Currently, most RKN are controlled by chemical agents, resulting in most nematicides facing problems such as reduced control effectiveness and pest resistance. In order to achieve the efficient and safe management of RKN, this test used copper sulfate and DMDS mixtures with CuSO₄ or NH₄HCO₃ to determine their toxicity against *M. incognita*. The combination of DMDS (2.5 mg/L) and CuSO₄ (46.58 mg/L) not only showed the best results with an $E - E_0$ value of 97.09% and a corrected mortality of 97.67%, but also has the lowest dosage. CuSO₄ mixed with other chemical nematicides also showed synergistic effects. CuSO₄ combined with deltamethrin and abamectin on *B. xylophilus* showed additive or synergistic effects [33]. Copper is an indispensable trace element for plant growth and plays a pivotal role, although the demand is small. In addition to mixing CuSO₄ as a trace fertilizer with pesticides, copper can also be used as an additive or nanomaterial and mixed into pesticides. Copper oxide nanopowders were mixed into cypermethrin to determine the toxic and synergistic effect on *Spodoptera litura* Fabricius and the results showed that the nanomaterials mixed with cypermethrin insecticide have synergistic effects.

Nanotechnology held great promise for mitigating the harmful effects of pesticides on the environment and human health, since it can provide systems enabling the controlled release of active compounds, thus increasing the efficiency and safety of products, while reducing the quantities required in field applications [40]. Therefore, further experiments are needed regarding the form of $CuSO_4$ mixed with DMDS for use in the field, alongside its significant synergistic effect indoors.

In the present study, the combination of DMDS (2.5 mg/L) and $NH_4HCO_3$ (80.25 mg/L) showed a synergistic effect on the control of *M. incognita*, with an $E - E_0$ value of 94.00% and a corrected mortality of 95.15%, and the usage is minimal. Similarly, the application of the ammonium sulfate and chitin in combination with the neem extracts reduced the root galling of *M. javanica* significantly [41]. A nanopesticide was formed by emamectin benzoate and glycine methyl ester was used as an organic nitrogen source. The biological experiments also showed that nanopesticides can maintain high insecticidal activity [42]. Furthermore, the data combined from 2019 and 2020 suggested that fluopyram seed treatment + $(NH4)_2 SO_4$ + Vydate + Max-In (R) Sulfur was effective at increasing seed cotton yields in the *Rotylenchulus reniformis* microplot trials. In *M. incognita* field trials, imidacloprid and thiodicarb + 28-0-0-5 + Vydate + Max-In (R) Sulfur supported the largest lint yields [43]. The integrated application of soil fumigation combined with fertigation and biocontrol agents could improve the RKN disease control efficacy further, demonstrated by reductions in diseases of 82.7–85.1% [44]. Therefore, the compound use of fumigant DMDS and nitrogen compounds has certain reference significance for practical production. DMDS, $CuSO_4$, and $NH_4HCO_3$ have nematocidal activities by themselves.

## 4. Conclusions

As chemical compounds, $CuSO_4$ and $NH_4HCO_3$ are absorbed by crops and they can control RKN. In this experiment, the results showed that $CuSO_4$ and $NH_4HCO_3$ had strong lethal activity on *M. incognita* J2 and the combination of DMDS and the two compounds had a synergistic effect. Although this experiment showed a good effect under indoor conditions, its practical application is limited by various factors such as different environments and soils. Therefore, further research is needed on the field control effect of these mixtures.

**Author Contributions:** Conceptualization, Q.W. (Qing Wang) and D.Y.; methodology, Z.S.; literature search, L.R.; software, W.F.; validation, Q.W. (Qiuxia Wang); formal analysis, Y.L.; investigation, Q.W. (Qing Wang); data curation, D.Y.; data interpretation, D.Z.; writing—original draft preparation, Q.W. (Qing Wang); writing—review and editing, D.Y.; visualization, A.C. All authors have read and agreed to the published version of the manuscript.

**Funding:** This research was funded by National Natural Science Foundation project of China (32172462), Beijing Talents Foundation (2018000021223ZK47).

**Institutional Review Board Statement:** Not applicable.

**Informed Consent Statement:** Not applicable.

**Data Availability Statement:** Not applicable.

**Conflicts of Interest:** The authors declare no conflict of interest.

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
