# Peer review of "Synergistic Activity of Dimethyl Disulfide Mixtures with Two Chemical Compounds against Meloidogyne incognita"

_sustainability, doi:10.3390/su142416711_

Round 1

Reviewer 1 Report

Please refer to attachment for comments

Reviewer 2 Report

Communication titled "Synergistic activity of dimethyl disulfide mixtures with two chemical compounds against Meloidogyne incognita" focused on a very fascinating idea and information that has many strong points, and is very well written & presented. Rectify some of the grammatical errors. I suggest you discuss and add some article information in the discussion section. Additionally, cross-reference all of the citations in the text with the references in the reference section. Make sure that all references have a corresponding source within the text and vice versa.

Minor comments are in the attached file. 

Reviewer 3 Report

Wang et al. investigated a new concept for the control of 21 root-knot nematode disease using synergistic effect of fumigant and 20 chemical compounds on M. incognita. The topic is of current interest, the manuscript entitled “Synergistic activity of dimethyl disulfide mixtures with two chemical compounds against Meloidogyne incognita” needs a substantial revision to be published. I have mentioned a few points of the manuscript that can be considered by the authors. 

Comment1: Line 20 says “corrected mortality” or improved mortality? Line 27 “cause damage in important crops” or “damage to”? line 54 “has” or have? Line 78 “to promote hatch of J2.” Or hatching? 

Comment2: line 58 “reduces the environmental pollution caused by the application of pesticides” how it “reduces” environmental pollution?

Comment3:  Under the sub heading “The Determination of toxicity of DMDS with CuSO4, NH4HCO3 on M. incognita J2” How many times was the treatment replicated? Line 96.

Comment4: Reframe the sentence “evaluate the effect of the true conveniently and effectively” Line-116.

Comment5: “Guo, M.; Zhang, W.; Li, J.; Liu, X., Research and practice on the interaction between pesticide and fertilizer. World 265 Agriculture 2000, (04), 39-41” This reference is not correctly cited. Line 265.

Comment6: “DNA of J2 was extracted according to the method of Feng et al 21 . Specific primers (MI-F: GTGAG- GATTCAGCTCCCCAG, MI-R: ACGAGGAACATACTTCTCCC)22 were used for PCR amplification.” Ref line 72 to 75. 

Comment7: Could you please mention the gene amplified and the product size. All the use of PCR i9n this study is not clear.

Comment8: Ref line 82-85 the stock concentration of CuSO4·5H2O is not given.

Comment9: Ref table no. 2 the no. of concentrations taken for different compounds is not same? Why so?

Comment10: “tset used copper sulfate” ref line- 181. Is it “test”??? please clarify.

Comment11: suggested to add few pictorial representations.
